# Evaluating the Financial Performances of the Publicly Held Healthcare Companies in Crisis Periods in Türkiye

**DOI:** 10.3390/healthcare11182588

**Published:** 2023-09-20

**Authors:** Dilaver Tengilimoğlu, Tolga Tümer, Russell L. Bennett, Mustafa Z. Younis

**Affiliations:** 1Department of Management, Faculty of Management, Atılım University, Gölbaşı, Ankara 06830, Türkiye; tolgatumer94@hotmail.com; 2Department of Health Policy and Management, College of Health Sciences, Jackson State University, Jackson, MS 39213, USA; russell.l.bennett@jsums.edu (R.L.B.); younis99@gmail.com (M.Z.Y.)

**Keywords:** healthcare companies, financial performance, crisis periods, economic crisis, COVID-19

## Abstract

The purpose of this study was to evaluate the financial performances of the publicly held healthcare companies in crisis periods in Türkiye. The 2018 economic crisis and the COVID-19 pandemic crisis were included in the study as the crisis periods. We collected the financial data of the publicly held healthcare companies and calculated three liquidity, three turnover, three leverage and three profitability ratios through ratio analysis to use as financial performance indicators. We then conducted Wilcoxon signed-rank tests and we performed separate analyses for the 2018 economic crisis and the COVID-19 pandemic crisis. The results of the analyses showed that there were no statistically significant differences between the publicly held healthcare companies’ liquidity, turnover, leverage, profitability ratios and thus their financial performances before the crises and after the crises. While the results are reassuring and give valuable insights to managers and policy makers to determine the areas that needs to be strengthened to be better prepared for possible future crises, our sample was limited. Therefore, this study presents an exploratory foundation for future studies which are needed to make a case for financial stability for the publicly held healthcare companies before and after the crisis periods.

## 1. Introduction

Crisis periods are times when the financial performances of firms are truly tested. In recent years, the biggest crisis that firms from all over the world had to face was the COVID-19 pandemic which caused the death of approximately seven million people worldwide according to the World Health Organization (WHO) [1]. As COVID-19 was spreading quickly and was also fatal, many countries had necessarily implemented lockdowns to slow down the spread of COVID-19. The necessary lockdown measures negatively and strongly affected the global economy [2,3]. According to the Organisation for Economic Co-operation and Development (OECD), real gross domestic product (GDP) growth for the world decreased by 3.5 percent in 2020, while there was an increase of 2.7 percent in 2019 and an average increase of 3.3 percent in 2013–2019; the unemployment rate for the world was 7.1 percent in 2020, while it was 5.4 in 2019 and its average was 6.5 in 2013–2019; world real trade growth decreased by 8.5 percent in 2020, while there was an increase of 1.3 percent in 2019 and an average increase of 3.4 percent in 2013–2019 [4]. The COVID-19 pandemic crisis is considered a sticky crisis in the literature, because it was complex, disrupted the global economy and therefore was harder to combat than a typical crisis [5,6,7].

Every sector and every firm were inevitably affected by the COVID-19 pandemic in one way or another, directly or indirectly. However, some sectors were affected by the COVID-19 pandemic more than other sectors and the healthcare sector was undoubtedly one of them. The pressure on the healthcare sector during the COVID-19 pandemic was relatively higher than other sectors, not just because the healthcare sector was on the frontline during the COVID-19 pandemic, but also because the healthcare sector had the responsibility of finding an effective and lasting cure to COVID-19. Global spending on health as a share of global GDP increased to 10.89 percent in 2020, while it was 9.83 percent in 2019 and 9.68 percent in 2018 [8]. According to the WHO, 63 percent of the global health spending in 2020 was from government sources, 36 percent of it was from private sources including out-of-pocket (OOP) spending, and 0.2 percent of it was from external sources which indicates health spending funded by nondomestic sources such as external aid, grants and donations [9].

The COVID-19 pandemic negatively affected many countries’ already overburdened health systems and thus had a negative impact on health service deliveries. Consequently, medical supply management, facility utilization and health human resource management became even more challenging during the COVID-19 pandemic [10]. Shreffler et al. [11] included 37 studies in their review article and they found consistent reports of stress, anxiety and depressive symptoms in healthcare workers resulting from the COVID-19 pandemic. Moreover, lockdowns that were imposed during the COVID-19 pandemic affected the global pharmaceutical supply chain [12].

There are studies in the literature that examine the impact of the COVID-19 pandemic on the financial performances of firms in the healthcare sector and its sub-sectors as well. He et al. [13] found that the total margin of California hospitals was not negatively affected by the COVID-19 pandemic. Moreover, Gidwani and Damberg [14] examined the financial performances of the US hospitals during the COVID-19 pandemic and found that most of the US hospitals were financially healthy. Zheng et al. [15] state that the impact of the COVID-19 pandemic on the financial performances of firms in the medicine sector in China was in fact positive. However, Mahssouni et al. [16] state that the financial performances of Belgian pharmaceutical companies were negatively affected by the COVID-19 pandemic.

In this study, we specifically focused on the financial performances of the publicly held healthcare companies in crisis periods in Türkiye. The Ministry of Health, which was formed in 1920, is the ministry responsible for health policy making and providing healthcare in Türkiye. Primary health care in Türkiye is solely provided by the Ministry of Health, although secondary and tertiary care is also provided by many other public and private institutions and organizations, such as universities and the Turkish Red Crescent [17,18].

The healthcare sector of Türkiye was also inevitably significantly affected by the COVID-19 pandemic. Global economic recession was especially a major concern for firms’ financial cycles and stability [19]. On the other hand, medical supply management, facility utilization and health human resource management became challenging during the COVID-19 pandemic in Türkiye as well. Tengilimoğlu et al. [20] found that the fear of contaminating their families with the COVID-19 virus was the major cause of the anxiety or stress among healthcare workers in Türkiye. Additionally, Hacimusalar et al. [21] found that the healthcare workers’ hopelessness and state of anxiety levels were higher than non-healthcare workers in Türkiye, which is understandable as the healthcare workers were in the frontline during the COVID-19 pandemic. However, Öncü et al. [22] analyzed the effect of the COVID-19 pandemic on health management and health services in Türkiye and found that Türkiye had managed the pandemic period sufficiently.

On the other hand, the COVID-19 pandemic was not the only crisis that the healthcare sector in Türkiye had to face in recent years. Just before the COVID-19 pandemic, an economic crisis had started in Türkiye in 2018 which is still going on today. The economy of Türkiye mainly depends on access to cheap credit sources and capital inflows for growth; in 2018, global financial conditions tightened and as a result, a currency crisis had started in Türkiye [23]. The distinctive features of the 2018 economic crisis are the depreciation of the Turkish Lira and the increase in inflation [24]. As with every other sector in the country, the healthcare sector was also inevitably affected by the economic crisis. Tülüce and Şafak [24] state that, although the 2018 economic crisis did not have a significant impact on health spending and health investments, the unemployment rates were higher.

In this context, this study aimed to evaluate the financial performances of the Turkish publicly held healthcare companies during the crisis periods of the 2018 economic crisis and the COVID-19 pandemic crisis. For this purpose, statistical analyses have been carried out. The evaluation of the financial performances of the publicly held healthcare companies in crisis periods can lay the foundation for deeper analyses to help policy makers and company managers determine the problematic areas that need to be strengthened to perform better in possible future crisis periods.

## 2. Materials and Methods

We carried out a case study of the publicly held healthcare companies in Türkiye and conducted statistical analyses to evaluate the financial performances of the publicly held healthcare companies in crisis periods in Türkiye. The significance level of 0.05 was used as the level of significance in the investigation. The crises that were included in the study, together with the years before crises and the years after crises are shown in Table 1.

The publicly held healthcare companies in Türkiye were determined using the “TradingView” [25] website. The publicly held healthcare companies in Türkiye with their codes, company names, sub-sectors and market capitalizations are presented in Table 2.

There are 15 publicly held healthcare companies in Türkiye as of 2023, but almost half of these companies went public only one or two years earlier and thus could not be included in the analyses because the required data for the crisis periods could not be collected. Accordingly, we collected the data of eight of these publicly held healthcare companies (AVHOL, DEVA, ECILC, LKMNH, MPARK, RTALB, SEYKM, TRILC). Among these companies, AVHOL, LKMNH and MPARK are in the health services sub-sector of the healthcare sector; DEVA, ECILC, RTALB and TRILC are in the technology services sub-sector of the healthcare sector. On the other hand, the market capitalizations of ECILC (TRY 33.345 billion), MPARK (TRY 24.756 billion) and DEVA (TRY 17.259 billion) are reasonably higher than the rest of the companies which all have market capitalizations below TRY 5.000 billion (AVHOL: TRY 4.358 billion, TRILC: TRY 2.228 billion, RTALB: TRY 1.663 billion, LKMNH: TRY 1.365 billion, SEYKM: TRY 0.785 billion).

In the study, the financial data of eight publicly held healthcare companies were collected from their annual balance sheet and income statement in independent audit reports. To measure the financial performances of firms, ratio analyses can be performed [14,26,27]. In this study, liquidity, turnover, leverage and profitability ratios were calculated to measure the financial performances of the publicly held healthcare companies. The financial ratios that were used in the study and their calculations are presented in Table 3.

Abbreviations of the financial ratios that were used in the study are also shown in Table 3. Other than the financial ratios, annual change in share prices were also included in the analyses and the required data were collected from the “Investing.com” [28] website. The letter ‘S’ is used to represent the annual change in share prices. The calculated values of the financial ratios and the annual changes in share prices are presented in Table 4.

After the calculation of the financial ratios and the annual changes in share prices, SPSS software was used to conduct statistical analyses. In the first step of the analyses, the distribution of quantitative variables for each crisis period that was included in the study was checked separately. For this purpose, basic descriptive statistics were calculated together with Kolmogorov–Smirnov and Shapiro–Wilk tests [29] examining the normality of distribution. The descriptive statistics and the results of the Kolmogorov–Smirnov and Shapiro–Wilk tests for the 2018 economic crisis are collectively shown in Table 5.

The mean values for all liquidity ratios were lower after the crisis (2.45 for L1; 1.95 for L2; 0.82 for L3) than before the crisis (2.68 for L1; 2.16 for L2; 1.14 for L3). The mean values for two turnover ratios were higher after the crisis (−9.60 for T1; 3.24 for T2) than before the crisis (−12.00 for T1; 3.22 for T2) and were the same for one turnover ratio after the crisis (0.66 for T3) and before the crisis (0.66 for T3). The mean values for two leverage ratios were lower after the crisis (0.44 for LV1; 0.25 for LV2) than before the crisis (0.49 for LV1; 0.32 for LV2) and were higher for one leverage ratio after the crisis (−3.41 for LV3) than before the crisis (−6.46 for LV3). The mean values for two profitability ratios were higher after the crisis (0.06 for P2; 0.10 for P3) than before the crisis (−0.07 for P2; 0.09 for P3) and were the same for one profitability ratio after the crisis (0.04 for P1) and before the crisis (0.04 for P1). The mean values for the annual change in share prices were lower after the crisis (−8.01 for S) than before the crisis (86.41 for S).

The significance level of the Kolmogorov–Smirnov and the Shapiro–Wilk statistic values shows whether the data are normally distributed; if the significance level is above 0.05, it means that the data are normally distributed [29]. Parametric tests should be used if the data are normally distributed and nonparametric tests should be used if the data are not normally distributed. If some of the data are normally distributed and some are not normally distributed, nonparametric tests would be more suitable.

According to the Kolmogorov–Smirnov statistic values and their significance levels, the data were not normally distributed before the crisis for all liquidity ratios (0.364 and *p* < 0.05 for L1; 0.374 and *p* < 0.05 for L2; 0.375 and *p* < 0.05 for L3), one of the leverage ratios (0.366 and *p* < 0.05 for LV3), one of the profitability ratios (0.441 and *p* < 0.05 for P2) and the annual change in share prices (0.415 and *p* < 0.05); the data were normally distributed for all turnover ratios (0.245 and *p* > 0.05 for T1; 0.217 and *p* > 0.05 for T2; 0.137 and *p* > 0.05 for T3), two of the leverage ratios (0.245 and *p* > 0.05 for LV1; 0.207 and *p* > 0.05 for LV2) and two of the profitability ratios (0.181 and *p* > 0.05 for P1; 0.183 and *p* > 0.05 for P3). The Kolmogorov–Smirnov statistic values and their significance levels also showed that the data were not normally distributed after the crisis for one of the liquidity ratios (0.330 and *p* < 0.05 for L3), one of the turnover ratios (0.304 and *p* < 0.05 for T1), one of the leverage ratios (0.379 and *p* < 0.05 for LV3) and the annual change in share prices (0.459 and *p* < 0.05); the data were normally distributed for two of the liquidity ratios (0.257 and *p* > 0.05 for L1; 0.258 and *p* > 0.05 for L2), two of the turnover ratios (0.180 and *p* > 0.05 for T2; 0.276 and *p* > 0.05 for T3), two of the leverage ratios (0.237 and *p* > 0.05 for LV1; 0.169 and *p* > 0.05 for LV2) and all profitability ratios (0.133 and *p* > 0.05 for P1; 0.208 and *p* > 0.05 for P2; 0.195 and *p* > 0.05 P3).

Additionally, according to the Shapiro–Wilk statistic values and their significance levels, the data were not normally distributed before the crisis for all liquidity ratios (0.707 and *p* < 0.05 for L1; 0.731 and *p* < 0.05 for L2; 0.745 and *p* < 0.05 for L3), one of the leverage ratios (0.732 and *p* < 0.05 for LV3), one of the profitability ratios (0.600 and *p* < 0.05 for P2) and the annual change in share prices (0.681 and *p* < 0.05 for S); the data were normally distributed for all turnover ratios (0.833 and *p* > 0.05 for T1; 0.895 and *p* > 0.05 for T2; 0.971 and *p* > 0.05 for T3), two of the leverage ratios (0.882 and *p* > 0.05 for LV1; 0.907 and *p* > 0.05 for LV2) and two of the profitability ratios (0.944 and *p* > 0.05 for P1; 0.969 and *p* > 0.05 for P3). The Shapiro–Wilk statistic values and their significance levels also showed that the data were not normally distributed after the crisis for one of the liquidity ratios (0.790 and *p* < 0.05 for L3), one of the turnover ratios (0.779 and *p* < 0.05 for T1), one of the leverage ratios (0.745 and *p* < 0.05 for LV3) and the annual change in share prices (0.585 and *p* < 0.05 for S); the data were normally distributed for two of the liquidity ratios (0.847 and *p* > 0.05 for L1; 0.845 and *p* > 0.05 for L2), two of the turnover ratios (0.926 and *p* > 0.05 for T2; 0.842 and *p* > 0.05 for T3), two of the leverage ratios (0.884 and *p* > 0.05 for LV1; 0.933 and *p* > 0.05 for LV2) and all profitability ratios (0.979 and *p* > 0.05 for P1; 0.925 and *p* > 0.05 for P2; 0.903 and *p* > 0.05 for P3). The descriptive statistics and the results of the Kolmogorov–Smirnov and Shapiro–Wilk tests for the COVID-19 pandemic are collectively shown in Table 6.

The mean values for two liquidity ratios were lower after the crisis (1.80 for L1; 1.51 for L2) than before the crisis (2.05 for L1; 1.72 for L2) and were higher for one liquidity ratio after the crisis (0.86 for L3) than before the crisis (0.80 for L3). The mean values for all turnover ratios were higher after the crisis (−7.69 for T1; 3.21 for T2; 0.60 for T3) than before the crisis (−9.11 for T1; 3.15 for T2; 0.57 for T3). The mean values for two leverage ratios were lower after the crisis (0.25 for LV2; −6.63 for LV3) than before the crisis (0.30 for LV2; −5.76 for LV3) and were higher for one leverage ratio after the crisis (0.49 for LV1) than before the crisis (0.48 for LV1). The mean values for all profitability ratios were higher after the crisis (0.12 for P1; 0.23 for P2; 0.20 for P3) than before the crisis (0.06 for P1; 0.14 for P2; 0.15 for P3). The mean values for the annual change in share prices were higher after the crisis (325.16 for S) than before the crisis (55.45 for S).

According to the Kolmogorov–Smirnov statistic values and their significance levels, the data were not normally distributed before the crisis for one of the leverage ratios (0.428 and *p* < 0.05 for LV3); the data were normally distributed for all liquidity ratios (0.241 and *p* > 0.05 for L1; 0.204 and *p* > 0.05 for L2; 0.303 and *p* > 0.05 for L3), all turnover ratios (0.279 and *p* > 0.05 for T1; 0.188 and *p* > 0.05 for T2; 0.226 and *p* > 0.05 for T3), two of the leverage ratios (0.195 and *p* > 0.05 for LV1; 0.282 and *p* > 0.05 for LV2), all profitability ratios (0.226 and *p* > 0.05 for P1; 0.155 and *p* > 0.05 for P2; 0.163 and *p* > 0.05 for P3) and the annual change in share prices (0.230 and *p* > 0.05 for S). The Kolmogorov–Smirnov statistic values and their significance levels also showed that the data were not normally distributed after the crisis for one of the turnover ratios (0.373 and *p* < 0.05 for T1); the data were normally distributed for all liquidity ratios (0.298 and *p* > 0.05 for L1; 0.299 and *p* > 0.05 for L2; 0.293 and *p* > 0.05 for L3), two of the turnover ratios (0.153 and *p* > 0.05 for T2; 0.168 and *p* > 0.05 for T3), all leverage ratios (0.138 and *p* > 0.05 for LV1; 0.301 and *p* > 0.05 for LV2; 0.250 and *p* > 0.05 for LV3), all profitability ratios (0.284 and *p* > 0.05 for P1; 0.171 and *p* > 0.05 for P2; 0.257 and *p* > 0.05 for P3) and the annual change in share prices (0.277 and *p* > 0.05 for S).

Additionally, according to the Shapiro–Wilk statistic values and their significance levels, the data were not normally distributed before the crisis for one of the liquidity ratios (0.800 and *p* < 0.05 for L3), one of the turnover ratios (0.763 and *p* < 0.05 for T1) and one of the leverage ratios (0.561 and *p* < 0.05 for LV3); the data were normally distributed for two of the liquidity ratios (0.867 and *p* > 0.05 for L1; 0.918 and *p* > 0.05 for L2), two of the turnover ratios (0.928 and *p* > 0.05 for T2; 0.866 and *p* > 0.05 for T3), two of the leverage ratios (0.959 and *p* > 0.05 for LV1; 0.891 and *p* > 0.05 for LV2), all profitability ratios (0.848 and *p* > 0.05 for P1; 0.981 and *p* > 0.05 for P2; 0.942 and *p* > 0.05 for P3) and the annual change in share prices (0.905 and *p* > 0.05 for S). The Shapiro–Wilk statistic values and their significance levels also showed that the data were not normally distributed after the crisis for two of the liquidity ratios (0.797 and *p* < 0.05 for L2; 0.771 and *p* < 0.05 for L3), one of the turnover ratios (0.719 and *p* < 0.05 for T1), one of the leverage ratios (0.790 and *p* < 0.05 for LV3) and the annual change in share prices (0.737 and *p* < 0.05 for S); the data were normally distributed for one of the liquidity ratios (0.831 and *p* > 0.05 for L1), two of the turnover ratios (0.929 and *p* > 0.05 for T2; 0.929 and *p* > 0.05 for T3), two of the leverage ratios (0.988 and *p* > 0.05 for LV1; 0.850 and *p* > 0.05 for LV2) and all profitability ratios (0.854 and *p* > 0.05 for P1; 0.927 and *p* > 0.05 for P2; 0.853 and *p* > 0.05 for P3).

After examining the descriptive statistics and the results of the normality tests for each crisis period, it was determined that the Wilcoxon signed-rank test was the most suitable statistical analysis for the data set. The Wilcoxon signed-rank test is a nonparametric test which can be used to compare two related samples to assess whether their mean ranks differ [30,31]. Therefore, Wilcoxon signed-rank tests were conducted for each crisis period that was included in the study using the calculated values of the financial ratios of the publicly held healthcare companies. SPSS software was used for the analyses.

## 3. Results

In the study, Wilcoxon signed-rank tests were conducted using the financial data of the publicly held healthcare companies in Türkiye. Analyses were conducted for the 2018 economic crisis and the COVID-19 pandemic separately. The Wilcoxon signed-rank test results for the 2018 Economic Crisis are shown in Table 7.

Negative Ranks show the number of publicly held healthcare companies that had a lower value for their financial performance indicator after the crisis than before the crisis. Conversely, Positive Ranks show the number of publicly held healthcare companies that had a higher value for their financial performance indicator after the crisis than before the crisis. On the other hand, Ties show the number of publicly held healthcare companies that had the same value for their financial performance indicator after the crisis and before the crisis [30,31]. Accordingly, the values of all liquidity ratios were higher after the crisis for more than half of the publicly held healthcare companies (Negative Ranks: 3, Positive Ranks: 5, Ties: 0 for L1, L2, L3). The values of two turnover ratios were higher after the crisis for half of the publicly held healthcare companies and were lower for the other half of the publicly held healthcare companies (Negative Ranks: 4, Positive Ranks: 4, Ties: 0 for T1, T2). While the values of one turnover ratio were higher after the crisis for half of the publicly held healthcare companies, they were lower for three companies and were the same for one company (Negative Ranks: 3, Positive Ranks: 4, Ties: 1 for T3). The values of two leverage ratios were higher after the crisis for more than half of the publicly held healthcare companies (Negative Ranks: 2, Positive Ranks: 5, Ties: 1 for LV1; Negative Ranks: 3, Positive Ranks: 5, Ties: 0 for LV3). However, the values of one leverage ratio were lower after the crisis for half of the publicly held healthcare companies, were higher for two of the publicly held healthcare companies and were the same for two of the publicly held healthcare companies (Negative Ranks: 4, Positive Ranks: 2, Ties: 2 for LV2). The values of two profitability ratios were higher after the crisis for more than half of the publicly held healthcare companies (Negative Ranks: 1, Positive Ranks: 5, Ties: 2 for P1; Negative Ranks: 2, Positive Ranks: 6, Ties: 0 for P2), but the values of one profitability ratio were lower for two of the publicly held healthcare companies, were higher for three of the publicly held healthcare companies and were the same for three of the publicly held healthcare companies (Negative Ranks: 2, Positive Ranks: 3, Ties: 3 for P3). Finally, the values of the annual change in share prices were lower after the crisis for the majority of the publicly held healthcare companies (Negative Ranks: 5, Positive Ranks: 1, Ties 0 for S).

The Z statistic shows the value of the Wilcoxon signed-rank test statistic and if the significance level (p) of the Z statistic is higher than 0.05, it means that there is no significant difference between the compared values. Accordingly, there were no significant differences between the liquidity ratios’ values before the crisis and after the crisis (Z: −0.140 and *p* > 0.05 for L1; Z: −0.140 and *p* > 0.05 for L2; Z: −0.140 and *p* > 0.05 for L3); between the turnover ratios’ values before the crisis and after the crisis (Z: −0.911 and *p* > 0.05 for T1; Z: −0.280 and *p* > 0.05 for T2; Z: −0.423 and *p* > 0.05 for T3); between the leverage ratios’ values before the crisis and after the crisis (Z: −0.169 and *p* > 0.05 for LV1; Z: −1.572 and *p* > 0.05 for LV2; Z: −1.680 and *p* > 0.05 for LV3); between the profitability ratios’ values before the crisis and after the crisis (Z: −0.970 and *p* > 0.05 for P1; Z: −1.404 and *p* > 0.05 for P2; Z: −0.412 and *p* > 0.05 for P3); and between the annual change in share prices’ values before the crisis and after the crisis (Z: −1.153 and *p* > 0.05 for S).

According to the analyses’ results, there were no significant differences between any of the financial performance indicators’ values in the year before the crisis and the values in the year after the crisis for the 2018 economic crisis. The Wilcoxon signed-rank test results for the COVID-19 pandemic are presented in Table 8.

The values of all liquidity ratios were higher after the crisis for more than half of the publicly held healthcare companies (Negative Ranks: 3, Positive Ranks: 5, Ties: 0 for L1, L2, L3). The values of one turnover ratio were higher after the crisis for more than half of the publicly held healthcare companies (Negative Ranks: 3, Positive Ranks: 5, Ties: 0 for T1), while the values of one turnover ratio were lower after the crisis for more than half of the publicly held healthcare companies (Negative Ranks: 6, Positive Ranks: 2, Ties: 0 for T3). Moreover, the values of one turnover ratio were lower after the crisis for half of the publicly held healthcare companies, were higher for three of the publicly held healthcare companies and were the same for one of the publicly held healthcare companies (Negative Ranks: 4, Positive Ranks: 3, Ties: 1 for T2). The values of two leverage ratios were lower after the crisis for more than half of the publicly held healthcare companies (Negative Ranks: 5, Positive Ranks: 1, Ties: 2 for LV2; Negative Ranks: 5, Positive Ranks: 3, Ties: 0 for LV3). While the values of one leverage ratio were lower after the crisis for half of the publicly held healthcare companies, they were higher for three of the publicly held healthcare companies and were the same for one of the publicly held healthcare companies (Negative Ranks: 4, Positive Ranks: 3, Ties: 1 for LV1). The values of all profitability ratios were higher after the crisis for the majority of the publicly held healthcare companies (Negative Ranks: 1, Positive Ranks: 6, Ties: 1 for P1, P3; Negative Ranks: 1, Positive Ranks: 7, Ties: 0 for P2). Finally, the values of the annual change in share prices were higher after the crisis for the majority of the publicly held healthcare companies (Negative Ranks: 2, Positive Ranks: 5, Ties 0 for S).

There were no significant differences between the liquidity ratios’ values before the crisis and after the crisis (Z: −0.280 and *p* > 0.05 for L1; Z: −0.421 and *p* > 0.05 for L2; Z: −0.631 and *p* > 0.05 for L3); between the turnover ratios’ values before the crisis and after the crisis (Z: −1.120 and *p* > 0.05 for T1; Z: −0.000 and *p* > 0.05 for T2; Z: −1.193 and *p* > 0.05 for T3); between the leverage ratios’ values before the crisis and after the crisis (Z: −0.085 and *p* > 0.05 for LV1; Z: −1.156 and *p* > 0.05 for LV2; Z: −0.561 and *p* > 0.05 for LV3); between the profitability ratios’ values before the crisis and after the crisis (Z: −1.690 and *p* > 0.05 for P1; Z: −1.895 and *p* > 0.05 for P2; Z: −1.183 and *p* > 0.05 for P3); and between the annual change in share prices’ values before the crisis and after the crisis (Z: −1.859 and *p* > 0.05 for S).

According to the analyses’ results, there were no significant differences between any of the financial performance indicators’ values in the year before the crisis and the values in the year after the crisis for the COVID-19 pandemic.

## 4. Discussion

The financial performances of firms are always at risk of deteriorating, but this risk greatly increases during crisis periods such as economic crises and pandemics. Major crises generally affect every sector in varying ways and degrees, and healthcare sector is no different. In fact, the COVID-19 pandemic, which is thought to be the biggest crisis the world had experienced in a long time, affected the healthcare sector more than other sectors because the healthcare sector was both on the frontline and responsible for finding a cure during the COVID-19 pandemic.

In this context, we statistically analyzed the financial performances of the publicly held healthcare companies in crisis periods in Türkiye. We included the 2018 economic crisis and the COVID-19 pandemic crisis as the crisis periods. We collected the financial data of the publicly held healthcare companies and we performed a ratio analysis to calculate three liquidity, three turnover, three leverage and three profitability ratios. Then, we conducted Kolmogorov–Smirnov and Shapiro–Wilk normality tests and examined the results together with the descriptive statistics of the data. It was determined that conducting the nonparametric Wilcoxon signed-rank test would be the most suitable.

The results of the Wilcoxon signed-rank tests showed that the liquidity, turnover, leverage and profitability ratios of the publicly held healthcare companies and thus their financial performances after the crisis periods were not significantly different from their liquidity, turnover, leverage and profitability ratios and thus their financial performances before the crisis periods. The findings of the study are in accordance with the findings of other similar studies that found no negative impact of the COVID-19 pandemic on the financial performances of healthcare companies [13,14,15]. However, there are also studies that found a negative impact of the COVID-19 pandemic on healthcare companies’ financial performances [16].

Even though there are studies that showed that the healthcare sector in Türkiye was negatively affected by the COVID-19 pandemic [20,21], we found that the publicly held healthcare companies in Türkiye were able to keep their liquidity, turnover, leverage and profitability ratios steady. The findings of the study are concordant with Öncü et al.’s [22] findings. Moreover, despite the fact that unemployment rates were higher during the 2018 economic crisis [24], this did not have a significant impact on the publicly held healthcare companies’ liquidity, turnover, leverage and profitability ratios according to the results of our analyses.

Managers and policy makers may benefit from our analyses to examine and determine the areas that need to be strengthened in order to be better prepared for possible future crisis periods. However, the study has some limitations. The main limitation of the study is the fact that any case for financial stability before and after the crisis periods cannot be made based on the evidence, especially because the sample size is very small. Similar analyses could be conducted using different financial performance criteria and statistical methods to collect data from a bigger sample. Another limitation of the study is that we only included the 2018 economic crisis and the COVID-19 pandemic crisis as the crisis periods. Similar analyses could be conducted over a longer time period to include more crisis periods. Moreover, we only focused on the Turkish publicly held healthcare companies. Similar analyses could also be conducted in other countries.

## Figures and Tables

**Table 1 healthcare-11-02588-t001:** The crises that were included in the study, the years before crises and the years after crises for Wilcoxon signed-rank tests.

Crises	Year before Crisis	Year after Crisis
2018 Economic Crisis	2017	2018
COVID-19 Pandemic	2019	2020

**Table 2 healthcare-11-02588-t002:** The codes, the company names, the sub-sectors and the market capitalizations of the publicly held healthcare companies in Türkiye.

Codes	Company Names	Sub-Sectors	Market Capitalizations (Billion Turkish Liras)
ANGEN	ANATOLİA TANI VE BİYOTEKNOLOJİ ÜRÜNLERİ ARAŞTIRMA GELİŞTİRME SANAYİ VE TİCARET A.Ş.	Health Technology	4.191
AVHOL *	AVRUPA YATIRIM HOLDİNG A.Ş.	Health Services	4.358
DEVA *	DEVA HOLDİNG A.Ş.	Health Technology	17.259
ECILC *	EİS ECZACIBAŞI İLAÇ SINAİ VE FİNANSAL YATIRIMLAR SANAYİ VE TİCARET A.Ş.	Health Technology	33.345
EGEPO	NASMED ÖZEL SAĞLIK HİZMETLERİ TİCARET A.Ş.	Health Services	2.905
GENIL	GEN İLAÇ VE SAĞLIK ÜRÜNLERİ SANAYİ VE TİCARET A.Ş.	Health Technology	20.055
KAYSE	KAYSERİ ŞEKER FABRİKASI A.Ş.	Health Technology	No data
LKMNH *	LOKMAN HEKİM ENGÜRÜSAĞ SAĞLIK TURİZM EĞİTİM HİZMETLERİ VE İNŞAAT TAAHHÜT A.Ş.	Health Services	1.365
MEDTR	MEDİTERA TIBBİ MALZEME SANAYİ VE TİCARET A.Ş.	Health Technology	4.624
MPARK *	MLP SAĞLIK HİZMETLERİ A.Ş.	Health Services	24.756
ONCSM	ONCOSEM ONKOLOJİK SİSTEMLER SANAYİ VE TİCARET A.Ş.	Health Technology	No data
RTALB *	RTA LABORATUVARLARI BİYOLOJİK ÜRÜNLER İLAÇ VE MAKİNE SANAYİ TİCARET A.Ş.	Health Technology	1.663
SEYKM *	SEYİTLER KİMYA SANAYİ A.Ş.	Health Technology	0.785
TNZTP	TAPDİ OKSİJEN ÖZEL SAĞLIK VE EĞİTİM HİZMETLERİ SANAYİ TİCARET A.Ş.	Health Services	No data
TRILC *	TURK İLAÇ VE SERUM SANAYİ A.Ş.	Health Technology	2.228

* The companies that could be included in the analyses based on data availability.

**Table 3 healthcare-11-02588-t003:** The financial ratios that were used in the study and their calculations.

Financial Ratios		Calculations
Liquidity Ratios	L1: Current Ratio	Current Assets/Current Liabilities
	L2: Quick Ratio	(Current Assets—Inventories)/Current Liabilities
	L3: Cash Ratio	(Current Assets—Inventories—Accounts Receivable)/Current Liabilities
Turnover Ratios	T1: Inventory Turnover Ratio	Cost of Goods Sold/Inventories
	T2: Accounts Receivable Turnover Ratio	Net Sales/Accounts Receivable
	T3: Asset Turnover Ratio	Net Sales/Total Assets
Leverage Ratios	LV1: Total Debt Ratio	Total Debt/Total Assets
	LV2: Long-term Debt Ratio	Long-term Debt/Total Assets
	LV3: Interest Coverage Ratio	Earnings Before Interest and Taxes/Interest Expense
Profitability Ratios	P1: Return on Assets	Net Profit/Total Assets
	P2: Return on Equity	Net Profit/Equity
	P3: Net Profit Margin	Net Profit/Net Sales

**Table 4 healthcare-11-02588-t004:** The values of the financial ratios and the annual changes in share prices.

Years	Firms	L1	L2	L3	T1	T2	T3	LV1	LV2	LV3	P1	P2	P3	S
2017	AVHOL	0.70	0.66	0.02	−20.04	1.59	0.64	0.80	0.41	0.15	0.00	0.01	0.00	−6.15
	DEVA	1.48	0.97	0.27	−2.11	2.74	0.64	0.52	0.28	−1.57	0.07	0.15	0.12	24.43
	ECILC	3.53	3.18	2.49	−5.00	4.10	0.16	0.09	0.04	−28.59	0.04	0.04	0.24	60.03
	LKMNH	1.00	0.85	0.17	−16.61	4.41	1.06	0.66	0.48	−0.59	0.03	0.08	0.03	58.68
	MPARK	0.92	0.88	0.27	−42.27	3.43	0.95	0.96	0.93	0.45	−0.05	−1.17	−0.05	No data
	RTALB	4.94	4.13	2.33	−3.81	2.67	0.61	0.14	0.01	−7.44	0.07	0.08	0.11	38.75
	SEYKM	8.02	5.85	3.46	−2.97	4.60	0.75	0.11	0.03	−13.10	0.08	0.09	0.11	342.72
	TRILC	0.86	0.74	0.12	−3.14	2.24	0.43	0.61	0.35	−0.99	0.07	0.17	0.16	No data
2018	AVHOL	2.09	1.84	0.42	−12.63	2.61	0.82	0.42	0.25	0.09	0.00	0.00	0.00	99.18
	DEVA	1.65	1.00	0.31	−1.38	2.58	0.62	0.55	0.31	−1.28	0.08	0.18	0.13	−24.89
	ECILC	3.77	3.41	2.64	−5.11	3.72	0.16	0.10	0.05	−16.67	0.05	0.06	0.35	−34.69
	LKMNH	0.71	0.56	0.10	−11.63	4.65	1.00	0.71	0.41	−0.71	0.03	0.10	0.03	−25.35
	MPARK	0.98	0.93	0.28	−32.97	3.48	0.97	0.82	0.67	0.35	−0.04	−0.22	−0.04	No data
	RTALB	3.79	3.35	1.13	−4.54	2.45	0.31	0.14	0.01	2.91	0.01	0.01	0.03	−36.94
	SEYKM	5.69	3.64	1.51	−3.25	4.67	0.91	0.13	0.03	−11.45	0.10	0.12	0.11	−25.38
	TRILC	0.95	0.87	0.15	−5.29	1.77	0.51	0.67	0.29	−0.54	0.08	0.23	0.15	No data
2019	AVHOL	1.72	1.59	0.33	−8.50	1.00	0.22	0.40	0.27	−1.06	0.06	0.10	0.27	−36.63
	DEVA	1.85	1.21	0.49	−1.52	2.85	0.65	0.51	0.29	−1.89	0.13	0.28	0.21	173.28
	ECILC	2.79	2.55	1.89	−6.34	3.76	0.18	0.12	0.05	−2.49	0.03	0.04	0.19	52.95
	LKMNH	0.79	0.62	0.14	−11.82	4.88	0.90	0.75	0.58	−0.14	0.02	0.09	0.02	29.83
	MPARK	0.89	0.84	0.30	−31.52	3.73	0.95	0.94	0.88	−0.14	0.01	0.15	0.01	49.53
	RTALB	4.43	4.05	1.96	−4.21	2.37	0.17	0.12	0.04	−19.33	0.05	0.05	0.26	42.96
	SEYKM	3.03	2.07	1.20	−2.82	5.11	0.89	0.25	0.03	−20.52	0.13	0.17	0.15	76.34
	TRILC	0.89	0.81	0.12	−6.11	1.53	0.57	0.76	0.29	−0.48	0.05	0.21	0.09	No data
2020	AVHOL	0.98	0.87	0.23	−4.00	0.91	0.25	0.45	0.02	−0.16	0.00	0.00	0.00	289.61
	DEVA	1.97	1.39	0.69	−1.78	2.70	0.60	0.47	0.22	−5.59	0.21	0.40	0.35	169.10
	ECILC	3.06	2.70	2.16	−4.08	4.03	0.15	0.12	0.05	−7.98	0.05	0.06	0.35	98.11
	LKMNH	0.74	0.61	0.16	−12.53	4.94	0.81	0.67	0.47	−2.00	0.06	0.19	0.08	85.76
	MPARK	0.92	0.87	0.35	−26.95	3.47	0.88	0.92	0.84	−0.27	0.01	0.19	0.02	24.19
	RTALB	1.73	1.51	0.58	−6.65	3.12	0.90	0.35	0.04	−17.44	0.24	0.36	0.26	1201.14
	SEYKM	3.94	3.31	2.58	−3.68	5.11	0.74	0.33	0.16	−18.63	0.28	0.42	0.38	408.20
	TRILC	1.06	0.84	0.09	−2.12	1.39	0.50	0.63	0.22	−0.96	0.08	0.23	0.17	No data

**Table 5 healthcare-11-02588-t005:** The descriptive statistics and the results of the Kolmogorov–Smirnov and Shapiro–Wilk tests for the 2018 economic crisis.

Years	Firms	Descriptive Statistics					Kolmogorov–Smirnov			Shapiro–Wilk		
		N	Min	Max	Mean	Std. Dev.	Statistic	df	Sig. (p)	Statistic	df	Sig. (p)
Before Crisis (2017)	L1	8	0.70	8.02	2.68	2.64	0.364	6	0.012	0.707	6	0.007
	L2	8	0.66	5.85	2.16	1.98	0.374	6	0.009	0.731	6	0.013
	L3	8	0.02	3.46	1.14	1.38	0.375	6	0.009	0.745	6	0.018
	T1	8	−42.27	−2.11	−12.00	14.02	0.245	6	0.200	0.833	6	0.113
	T2	8	1.59	4.60	3.22	1.09	0.217	6	0.200	0.895	6	0.346
	T3	8	0.16	1.06	0.66	0.28	0.137	6	0.200	0.971	6	0.899
	LV1	8	0.09	0.96	0.49	0.34	0.245	6	0.200	0.882	6	0.278
	LV2	8	0.01	0.93	0.32	0.31	0.207	6	0.200	0.907	6	0.414
	LV3	8	−28.59	0.45	−6.46	10.10	0.366	6	0.012	0.732	6	0.013
	P1	8	−0.05	0.08	0.04	0.04	0.181	6	0.200	0.944	6	0.692
	P2	8	−1.17	0.23	−0.07	0.45	0.441	6	0.001	0.600	6	0.000
	P3	8	−0.05	0.24	0.09	0.09	0.183	6	0.200	0.969	6	0.884
	S	6	−6.15	342.72	86.41	127.94	0.415	6	0.002	0.681	6	0.004
After Crisis (2018)	L1	8	0.71	5.69	2.45	1.78	0.257	6	0.200	0.847	6	0.149
	L2	8	0.56	3.64	1.95	1.31	0.258	6	0.200	0.845	6	0.143
	L3	8	0.10	2.64	0.82	0.89	0.330	6	0.040	0.790	6	0.048
	T1	8	−32.97	−1.38	−9.60	10.22	0.304	6	0.088	0.779	6	0.038
	T2	8	1.77	4.67	3.24	1.06	0.180	6	0.200	0.926	6	0.553
	T3	8	0.16	1.00	0.66	0.32	0.276	6	0.170	0.842	6	0.137
	LV1	8	0.10	0.82	0.44	0.29	0.237	6	0.200	0.884	6	0.286
	LV2	8	0.01	0.67	0.25	0.22	0.169	6	0.200	0.933	6	0.607
	LV3	8	−16.67	2.91	−3.41	6.83	0.379	6	0.007	0.745	6	0.018
	P1	8	−0.04	0.10	0.04	0.05	0.133	6	0.200	0.979	6	0.945
	P2	8	−0.22	0.23	0.06	0.14	0.208	6	0.200	0.925	6	0.542
	P3	8	−0.04	0.35	0.10	0.12	0.195	6	0.200	0.903	6	0.393
	S	6	−36.94	99.18	−8.01	52.77	0.459	6	0.000	0.585	6	0.000

**Table 6 healthcare-11-02588-t006:** The descriptive statistics and the results of the Kolmogorov–Smirnov and Shapiro–Wilk tests for the COVID-19 pandemic.

Years	Firms	Descriptive Statistics					Kolmogorov–Smirnov			Shapiro–Wilk		
		N	Min	Max	Mean	Std. Dev.	Statistic	df	Sig. (p)	Statistic	df	Sig. (p)
Before Crisis (2019)	L1	8	0.79	4.43	2.05	1.29	0.241	7	0.200	0.867	7	0.175
	L2	8	0.62	4.05	1.72	1.15	0.204	7	0.200	0.918	7	0.454
	L3	8	0.12	1.96	0.82	0.77	0.303	7	0.051	0.800	7	0.041
	T1	8	−31.52	−1.52	−9.11	9.62	0.279	7	0.108	0.763	7	0.017
	T2	8	1.00	5.11	3.15	1.49	0.188	7	0.200	0.928	7	0.535
	T3	8	0.17	0.95	0.57	0.34	0.226	7	0.200	0.866	7	0.171
	LV1	8	0.12	0.94	0.48	0.31	0.195	7	0.200	0.959	7	0.806
	LV2	8	0.03	0.88	0.30	0.30	0.282	7	0.097	0.891	7	0.278
	LV3	8	−20.52	−0.14	−5.76	8.79	0.428	7	0.000	0.561	7	0.000
	P1	8	0.01	0.13	0.06	0.05	0.226	7	0.200	0.848	7	0.117
	P2	8	0.04	0.28	0.14	0.08	0.155	7	0.200	0.981	7	0.963
	P3	8	0.01	0.27	0.15	0.10	0.163	7	0.200	0.942	7	0.657
	S	7	−36.63	173.28	55.45	62.79	0.230	7	0.200	0.905	7	0.362
After Crisis (2020)	L1	8	0.74	3.94	1.80	1.15	0.298	7	0.060	0.831	7	0.082
	L2	8	0.61	3.31	1.51	0.98	0.299	7	0.058	0.797	7	0.038
	L3	8	0.09	2.58	0.86	0.96	0.293	7	0.071	0.771	7	0.021
	T1	8	−26.95	−1.48	−7.69	8.52	0.373	7	0.004	0.719	7	0.006
	T2	8	0.91	5.11	3.21	1.52	0.153	7	0.200	0.929	7	0.545
	T3	8	0.15	0.90	0.60	0.28	0.168	7	0.200	0.929	7	0.542
	LV1	8	0.12	0.92	0.49	0.25	0.138	7	0.200	0.988	7	0.989
	LV2	8	0.02	0.84	0.25	0.28	0.301	7	0.054	0.850	7	0.124
	LV3	8	−18.63	−0.16	−6.63	7.56	0.250	7	0.200	0.790	7	0.033
	P1	8	0.00	0.28	0.12	0.11	0.284	7	0.092	0.854	7	0.133
	P2	8	0.00	0.42	0.23	0.15	0.171	7	0.200	0.927	7	0.528
	P3	8	0.00	0.38	0.20	0.16	0.257	7	0.179	0.853	7	0.131
	S	7	24.19	1201.14	325.16	408.11	0.277	7	0.114	0.737	7	0.009

**Table 7 healthcare-11-02588-t007:** The results of the Wilcoxon signed-rank tests for the 2018 economic crisis.

Values	L1	L2	L3	T1	T2	T3	LV1	LV2	LV3	P1	P2	P3	S
Negative Ranks	3	3	3	4	4	3	2	4	3	1	2	2	5
Positive Ranks	5	5	5	4	4	4	5	2	5	5	6	3	1
Ties	0	0	0	0	0	1	1	2	0	2	0	3	0
Z statistic	−0.140	−0.140	−0.140	−0.911	−0.280	−0.423	−0.169	−1.572	−1.680	−0.970	−1.404	−0.412	−1.153
Sig. (p)	0.889	0.889	0.889	0.362	0.779	0.672	0.866	0.116	0.093	0.332	0.160	0.680	0.249

**Table 8 healthcare-11-02588-t008:** The results of the Wilcoxon signed-rank tests for the COVID-19 pandemic.

Values	L1	L2	L3	T1	T2	T3	LV1	LV2	LV3	P1	P2	P3	S
Negative Ranks	3	3	3	3	4	6	4	5	5	1	1	1	2
Positive Ranks	5	5	5	5	3	2	3	1	3	6	7	6	5
Ties	0	0	0	0	1	0	1	2	0	1	0	1	0
Z statistic	−0.280	−0.421	−0.631	−1.120	0.000	−1.193	−0.085	−1.156	−0.561	−1.690	−1.895	−1.183	−1.859
Sig. (p)	0.779	0.674	0.528	0.263	1.000	0.233	0.933	0.248	0.575	0.091	0.058	0.237	0.063

## Data Availability

Publicly available datasets were analyzed in the study. The data for financial ratios can be found in the section “Investor Relations” at the publicly held healthcare companies’ own websites and the data for the annual changes in share prices can be found at https://www.investing.com (accessed on 26 April 2023).

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
