# Peer review of "Evaluating the Financial Performances of the Publicly Held Healthcare Companies in Crisis Periods in Türkiye"

_healthcare, 2023, doi:10.3390/healthcare11182588_

Round 1

Reviewer 1 Report

It is a very interesting subject that investigated the impact of the financial performance of public health in the period of crisis in Turkey - COVID, economic crisis, and earthquakes. Of its topic, this article is suited for publishing in the Healthcare journal.  

I would like to ask authors a few questions and give suggestions for their work.

By reading the article, I noticed that data for earthquakes were not yet available for analysis but the earthquake as the crisis is described in the methodology, and shortly after that it is stated that this analysis will not be done. Why did you decide to mention it in the work if the data wasn’t available at the time? Also, I wanted to ask, is there a confirmed date when data about earthquakes will be available so you can complete your investigation with all three important crises? I think that would be of great importance to have that data analyzed in the article.

The Introduction is focused on the effects of the COVID crisis, on anxiety, stress, and depression. Still, as this is not the main focus of this investigation, I would suggest adding references to articles dealing with economic indicators like % share of GDP from the government for healthcare, OOP spending, or similar to make an Introduction more in line with your investigation purpose.

The economic crisis isn’t even shortly written in the Introduction, I would suggest adding a paragraph about it.

Introducing the healthcare system of Turkey in a few sentences would be interesting; the organization of it or maybe short historical development.

The material and methods should have described the type of the study. Adding the significant importance value - p and writing what level of significance is used in the investigation.  

Table 2 contains the names of all publicly held companies but not all of them were included in the study. Following the text I cannot find the names of companies that are included, I suggest adding the names of them or marking them in the table.

Tables with years of crisis should be supplemented with additional words – before and after the crisis, for easier follow and comparison. Same comment for the tables with ‘’Z‘’, it should be added what it stands for.

How are positive and negative ranks calculated or they are collected from the same source? They haven’t been commented on in the results.

The Discussion part should be expanded, focusing on the previous analysis of the different crises in the earlier periods of time, if the newer is not available. Finding a crucial explanation of the reasons for how the crisis affected the healthcare system and suggesting some valuable options for the future is necessary.

Author Response

Thank you very much for taking the time to review this manuscript. Please find the detailed responses and the corresponding revisions/corrections highlighted/in track changes in the re-submitted files.

Reviewer 2 Report

This study reports on: Evaluating the Financial Performances of the Publicly Held 2 Healthcare Companies in Crisis Periods in Türkiye

Various aspects of the impact of Covid are discussed and studied, I ask that the authors
specifically address each of my comments in their response.
I hope my comments, observations, and suggestions will allow the authors to
improve the manuscript and work towards publication or before
ubmit somewhere else. Below, I include comments pointing toward some of the issues.

The research methodology

In the first step of the analysis, the distribution of quantitative variables should be checked. For this purpose, basic descriptive statistics should be calculated together with, for example, the Shapiro-Wilk test examining the normality of distribution. Has such an analysis been carried out?      

Why the Wilcoxon test was used was not explained, what program was used for statistical research?

References

The literature review is cursory, only 17 items. It is not shown whether the researchers deal with the conducted research area in other countries?

Comprehensive literature review are needed that authors can show the novelty of this research

Conslusion

Recommendations/indications for future research should be included in the
discourse or conclusion.

Author Response

(The authors gave the same response as above.)

Reviewer 3 Report

The manuscript would be greatly improved by a few things to include a measure of the various companies size relative to one another, for example, market capitalization.  While ratios allow for a uniform metric between two or more companies, are the companies similar in terms of strategy and organization? In other words are they all in a single business line or are some vertically integrated or diversified?  Reference is made that this is a 10 year study, but the dates begin at 2017 and only through 2023, about 7 years.  Finally, the very small sample is concerning especially given the suggestion that there is necessarily evidence of financial stability pre and post crisis.  The stability suggested by the study also begs the question how was it accomplished? Were there reductions in service or layoffs?  While the basic research  question is interesting, I do not think any case can be made based on the evidence presented. 

Adequate

Author Response

(The authors gave the same response as above.)

Round 2

Reviewer 1 Report

Dear Authors, 

I am happy to tell you that all my suggestions and comments were addressed sufficiently and your research improvement is highly noticeable. I have nothing to add; everything is clear and nicely presented.

Sincerely,

Reviewer 3 Report

The revisions are quite helpful to the reader. Thank you for the opportunity to review your research, it is very innovative.